

# Aerosol Indirect Effects on Cirrus Clouds Based on Global-Scale Airborne Observations and Machine Learning Models

Derek Ngo[1], Minghui Diao[1], Ryan J. Patnaude[1,2], Sarah Woods[3], Glenn Diskin[4]

[1]Department of Meteorology and Climate Science, San Jose State University, San Jose, CA, 95192, USA
[2]*Current affiliation:* Department of Atmospheric Science, Colorado State University, Fort Collins, CO, 80521, USA
[3]NSF National Centre for Atmospheric Research, Research Aviation Facility, Broomfield, CO, 80021, USA
[4]NASA Langley Research Centre, Hampton, VA, 23681, USA

*Correspondence to*: Minghui Diao (minghui.diao@sjsu.edu)

**Abstract.** Cirrus cloud formation and evolution are subject to the influences of thermodynamic and dynamic conditions and aerosol indirect effects (AIEs). This study developed near global-scale in-situ aircraft observational datasets based on 12 field campaigns that spanned from the polar regions to the tropics, from 2008 to 2016. Cirrus cloud microphysical properties were investigated at temperatures ≤ -40 °C, including ice water content (IWC), ice crystal number concentration (Ni), and number-weighted mean diameter (Di). Positive correlations between the fluctuations of ice microphysical properties and the

fluctuations of aerosol number concentrations for larger (> 500 nm) and smaller (> 100 nm) aerosols (i.e., $Na_{500}$ and $Na_{100}$, respectively) were found, with stronger AIE from larger aerosols than smaller ones. Machine learning (ML) models showed that using relative humidity with respect to ice (RHi) as a predictor significantly increases the accuracy of predicting cirrus occurrences compared with temperature, vertical velocity (*w*), and aerosol number concentrations. The ML predictions of IWC fluctuations showed higher accuracies when larger aerosols were used as an predictor compared with smaller aerosols,

indicating the stronger AIE from larger aerosols than smaller ones, even though their AIEs are more similar when predicting the occurrences of cirrus. It is also important to capture the spatial variabilities of large aerosols at smaller scales as well as those of smaller aerosols at coarser scales to accurately simulate IWC in cirrus. These results can be used to improve understanding of aerosol-cloud interactions and evaluate model parameterizations of cirrus cloud properties and processes.

## 1 Introduction

Cirrus clouds are the one of the most prominent cloud types with a wide spatial coverage over the Earth's surface by 20% – 40% (Sassen et al., 2009; Mace and Wrenn, 2013). They are located in the upper troposphere around 8 – 17 km and are therefore composed entirely of ice crystals (Lynch et al., 2002). This altitude range also shows large sensitivities of the atmospheric radiative forcing to the amount of water vapor and ice crystals (e.g., Solomon et al., 2010; Tan et al., 2016). They are the type of clouds that impose particular challenges for both in-situ and remote sensing observations, due to their thin,

patchy nature, their vertical locations at higher altitudes, and the large spatial heterogeneities of their microphysical properties.



Representing various properties of cirrus clouds in global climate models (GCMs) is also critical for accurate estimation of global radiation budget and future climate prediction. The macrophysical properties (e.g., spatial extent, vertical thickness of cloud layers) and microphysical properties (e.g., mass and number concentrations of ice crystals) of cirrus clouds both have the potential to alter the radiative budget (Liou, 1992) and cause a significant climate feedback (Zhou et al., 2014).

Determining whether ice nucleation occurs is a critical step for accurately representing the radiative effect of an atmospheric column. Changing clear-sky ice supersaturation into a cirrus cloud given the same amount of total water content can produce an average of 2.49 W m$^{-2}$ radiative effects at the top of the atmosphere, with increased net radiation at in-cloud conditions by a range of 0.56 to 7.19 W m$^{-2}$ (Tan et al., 2016). Two mechanisms contribute to ice crystal formation at lower temperatures (e.g., temperatures $\leq$ -40 °C), that is, homogeneous freezing and heterogeneous freezing. The former mechanism spontaneously
freezes dilute aerosol solutions into ice crystals without the assistance of ice nucleating particles (INPs) depending upon the temperature and water activity (Schneider et al., 2021), while the latter mechanism relies on INPs to initiate ice nucleation via freezing pathways such as immersion freezing. It is still contested whether deposition freezing acts as a possible heterogeneous freezing mechanism at the cirrus temperature range as a previous study indicated that deposition freezing may be pore condensation freezing (Marcolli, 2014; David et al., 2019). Aerosol indirect effect (AIE) is important for the formation of
clouds because aerosols may contribute to heterogeneous freezing by serving as INPs or contribute to homogeneous freezing. Freezing of liquid aerosol solutions via homogeneous freezing requires much higher thresholds of relative humidity with respect to ice (RHi) (e.g., Koop et al., 2000). Comparatively, INPs can facilitate ice nucleation at lower RHi thresholds, although only a few types of aerosols have the capability to serve as INPs (e.g., Kanji et al., 2017, 2019). Previous aircraft-based in-situ measurements frequently observed mineral dust and metallic particles inside ice residuals in the midlatitudinal
cirrus, indicating that these aerosols frequently act as INPs in the real atmosphere. Other aerosols that may not act as an INP at mixed-phase cloud temperatures (> -38 °C), such as sea salt, may become an effective INP at cirrus temperatures (Patnaude et al., 2021a, 2024). In addition, black carbon has been found to have large variations in its effectiveness acting as INPs associated with various morphological and chemical characteristics and may increase the effectiveness during the aging and coating processes (e.g., Ullrich et al., 2017; Mahrt et al., 2018, 2020). The contribution and competition between homogeneous
and heterogeneous freezing may vary with pressure levels, geographical locations and meteorological conditions (e.g., deep convection, synoptic-scale forcing, and gravity waves), and the global distributions of each mechanism are not fully resolved (Cziczo et al., 2013; Mitchell et al., 2018).

Quantification of AIE has been a difficult topic because aside from AIE, various factors such as thermodynamic and dynamic conditions also affect cirrus clouds (e.g., Schiller et al., 2008; Patnaude and Diao, 2020). Isolating and quantifying the
contributions of individual factors on cirrus microphysical properties remains a challenging task for observational studies of the real atmosphere where environmental conditions cannot be fully controlled. In addition, cirrus clouds can also have different origins, such as convective liquid origin and in-situ origin, therefore can be subject to different environmental influences in their evolution (Krämer et al. 2016; Luebke et al. 2016; Krämer et al. 2020). Previously, Patnaude and Diao (2020) showed the importance of isolating other factors such as temperature, RHi, and $w$ before quantifying AIE, since these



other factors often play a more significant role in affecting ice microphysical properties. The authors used a "delta-delta" method, which basically examined the correlations between the fluctuations of ice microphysical properties and the fluctuations of aerosol number concentrations (Na) at controlled thermodynamic and dynamic conditions. That study allowed comparisons between larger (> 500 nm) and smaller aerosols (> 100 nm) for their correlations with cirrus microphysical properties, with implications for the possible contributions of heterogeneous and homogeneous freezing, respectively.

However, that method does not allow a direct comparison among the effects of various factors and therefore cannot address the question of which factor(s) are more influential than the others for cirrus formation and the subsequent cloud properties. Another technical drawback of that previous study is the lack of investigation of the small ice crystals due to the limitation of the cloud probe being used. That limits the understanding of AIE on homogeneous freezing since homogeneous freezing often forms numerous relatively smaller ice particles compared with heterogeneous freezing based on box model simulations (e.g.,

Spichtinger and Cziczo, 2010). Because of these limitations, a large in-situ observational dataset that includes measurements of both smaller and larger ice crystals as well as a new method that allows quantification and comparison of each factor need to be developed.

The limited understanding of AIE on cirrus clouds also inhibits the development of accurate parameterizations of aerosol-cloud interactions in GCMs. In fact, large uncertainties still exist in the simulations of AIE on cirrus in GCMs. Previous studies

comparing climate model simulations against in-situ observations found an underestimation of AIE by the simulations of the National Center for Atmospheric Research (NCAR) Community Earth System Model version 2 (CESM2) / Community Atmosphere Model version 6 (CAM6) (Patnaude et al., 2021a). The AIE on cirrus clouds is particularly underestimated at the earlier evolution stage of cirrus clouds such as the nucleation and early growth phases (Maciel et al., 2023). Adding or reducing aerosols can further modify cirrus properties, such as the cirrus thinning scenario discussed in hypothetical geoengineering

simulations (e.g., Storelvmo et al., 2013; Storelvmo and Herger, 2014; Muri et al., 2014; Gasparini and Lohmann, 2016; Lohmann and Gasparini, 2017; Liu and Shi, 2021). But due to the complexity of the processes affecting cirrus formation and evolution, more observational evidence is needed to verify the current parameterizations used in GCM simulations (e.g., Gettelman and Morrison, 2015), as well as the emerging types of parameterizations related to ice nucleation in cirrus clouds (e.g., Kärcher, 2022; Barahona et al., 2024).

This study develops large aircraft-based datasets using in-situ observations from multiple flight campaigns with near global coverage. A new method is developed based on a ML approach to quantify variations of cirrus microphysical properties and five controlling factors – temperature, RHi, $w$, $Na_{500}$, and $Na_{100}$. A new metric is developed to quantify individual effects of these five factors under three separate topics: (1) How do these factors affect formation of cirrus clouds? (2) How do they affect cirrus microphysical properties, in terms of the fluctuations of IWC being lower or higher relative to the average values?

And (3) how do they affect the distributions of IWC in cirrus clouds as a function of temperature, RHi, and $w$? The sections are designed as follows. Section 2 describes the observational datasets, instrumentation, and the set-up of the ML models. Section 3 examines each of the three topics mentioned above, by quantifying and contrasting the role of individual factors under each topic. Section 4 provides the main summary of the findings and their implications for climate simulations.



## 2 Observational Datasets and Experimental Setup

### 2.1 In-situ observations and instrumentation


A dataset focusing on the cirrus cloud temperature range was developed in this study based on seven U.S. National Science Foundation Campaigns (NSF) and five National Aeronautics and Space Administration (NASA) flight campaigns. All data used in this study are constrained to temperatures ≤ -40 ℃, to eliminate the presence of supercooled water droplets. The seven NSF flight campaigns in alphabetical order include CONTRAST (Pan et al., 2017), NSF-DC3 (Barth et al., 2015), HIPPO

(Wofsy, 2011), ORCAS (Stephens et al., 2018), PREDICT (Montgomery et al., 2012), START08 (Pan et al., 2010), and TORERO (Volkamer et al., 2015). The five NASA campaigns include ATTREX-2014 (Jensen et al., 2017a; Woods et al., 2018), NASA-DC3 (Barth et al., 2015), MACPEX (Rollins et al., 2014), POSIDON (Jensen et al., 2017b), and SEAC⁴RS (Toon et al., 2016). The DC3 campaign was a coordinated flight campaign between NASA and NSF, thus we use the NSF-DC3 and NASA-DC3 to differentiate the two research aircraft platforms during that campaign. Specific details of these

campaigns such as name, acronym, time, and location are listed in Table 1. Information of cirrus observations such as flight hours, ranges of temperatures, altitudes, and pressures are also shown in that table. Previously, these field campaigns were also used in Maciel et al. (2023) for the analysis of various phases of cirrus evolution. By compiling observations from these flight campaigns, we aim to construct a near global-scale dataset covering wide latitudinal regions (87 °N to 75 °S) and longitudinal regions (128 °E to 180 °E and 37 °W to 180 °W). Global maps illustrating the entire flight tracks of each NASA and NSF

campaign are shown in Figure 1.

Because one main objective of this study is to examine the effects of key environmental conditions (such as temperature, RHi and *w*) on cirrus properties, a few other campaigns that targeted cirrus clouds were not included in the compiled dataset due to issues with water vapor or RHi measurements at the cirrus temperature range. For example, the US Department of Energy (DOE) ARM Small Particles in Cirrus (SPARTICUS) campaign provided targeted observations of cirrus clouds but had issues

with water vapor measurements. The Learjet research aircraft also participated in the SEAC⁴RS campaign but did not provide good quality water vapor measurements below -30 ℃ due to the limitations of a chilled mirror hygrometer onboard.

The seven flight campaigns funded by U.S. NSF were carried out exclusively by the NSF/NCAR High-Performance Instrumented Airborne Platform for Environmental Research (HIAPER) Gulfstream V (GV) aircraft. It is worth noting that these seven NSF flight campaigns were not specifically designed for cirrus cloud measurements. For example, HIPPO was

planned for a near pole-to-pole profiling of greenhouse gases, DC3 targeted deep convective outflows, PREDICT targeted tropical cyclones, and START08 targeted the airmass exchanges between the stratosphere and troposphere, etc. The cirrus observations were extracted from these field campaigns since the GV aircraft often reached the upper troposphere and lower stratosphere as part of their flight planning.

A list of key variables and the instruments used to derive them are also shown in Table 1. The key measurements include 1-

Hz observations of basic meteorological parameters such as temperature, pressure, water vapor, *w*, as well as measurements of cloud microphysical properties (i.e., IWC, Ni, and Di) and aerosol number concentrations. Onboard the NSF/NCAR GV




research aircraft, the Vertical Cavity Surface Emitting Laser (VCSEL) hygrometer was used to measure molecular number concentrations of water vapor (Zondlo et al., 2010). The Rosemount temperature probe was used to provide 1-Hz temperature observations. Two cloud probes were used for the NSF campaigns, i.e., the Fast 2-Dimensional Cloud (Fast-2DC) probe and

the Cloud Droplet Probe (CDP). The CDP has a size range from 2 – 50 µm. The Fast-2DC has a physical measurement range of 62.5 – 1600 µm through a 64-photodiode array with 25-µm bin widths and mathematically reconstructs partially detected particles with the maximum size up to 3200 µm. The Fast-2DC probe was equipped with anti-shattering tips, and a post data processing has been applied through an "interarrival time rejection" algorithm, which is described in Field et al. (2006), although complete elimination of shattering was not possible for the current measurement technique especially for ice particles

smaller than 100 µm (e.g., Korolev et al. 2013). Measurements of aerosol number concentrations were obtained from the Ultra-High Sensitivity Aerosol Spectrometer (UHSAS), operating at a size range of 60 – 1000 nm with 99 logarithmically spaced bins.

In contrast to the NSF campaigns, the five NASA flight campaigns were obtained from several research aircraft platforms, including the NASA Global Hawk for ATTREX-2014, NASA DC-8 for SEAC$^4$RS and NASA-DC3, and NASA WB-57 for

MACPEX and POSIDON. The ATTREX, POSIDON and MACPEX were designed to sample cirrus clouds and advance the understanding of cirrus cloud microphysical properties, while the SEAC$^4$RS and NASA-DC3 campaigns were designed to target evolution of gases and aerosols in deep convective outflows. Compared with the other research aircraft platforms that mostly sampled altitudes lower than 15 km, the ATTREX and POSIDON campaigns sampled mostly above 15 km onboard the NASA Global Hawk aircraft and NASA WB-57. The ATTREX campaign had four deployments between 2011 and 2015,

and only the 2014 deployment is used in the compiled dataset based on the availability of both ice microphysical properties and water vapor measurements.

Water vapor measurements during ATTREX, POSIDON, DC3, and SEAC$^4$RS campaigns were obtained from the Diode Laser Hygrometer (DLH), which operates at a near-infrared wavelength of 1.4 µm. The water vapor measurements in MACPEX were sampled using the Harvard Water Vapor (HWV) instrument, which is a combination of measurement methodologies

from the Lyman-α photo-fragment fluorescence instrument (LyA) and Harvard Herriott Hygrometer (HHH). Temperature measurements were based on the NASA Meteorological Measurement System (MMS) onboard various research aircraft. For all the NSF and NASA campaigns, saturation pressures with respect to ice (e$_s$) were derived from temperature measurements based on the equation from Murphy and Koop (2005), which were further combined with water vapor measurements to calculate RHi. Aerosol measurements were provided in three NASA campaigns (i.e., MACPEX, DC3, and SEAC$^4$RS). DC3

and SEAC$^4$RS utilized UHSAS, similar to NSF campaigns, while MACPEX used the Focused Cavity Aerosol Spectrometer (FCAS) that measures particles within the diameter range of 70 – 1000 µm. The NASA ATTREX and POSIDON campaigns were not included in the analysis of AIE due to the lack of aerosol measurements.

Ice particle measurements for most of the five NASA campaigns were based on two probes – the Fast-CDP probe and the Two-Dimensional Stereo Probe (2DS). The Fast-CDP (FCDP) probe has a size range of 1 – 50 µm. The 2DS probe has a

dynamic range of 5 – 3005 µm and uses two linear and independent 128-photodiode arrays designed to record at a 10 µm pixel



resolution. Similar to the Fast-2DC probe in the NSF campaigns, the 2DS probe also installed anti-shattering tips for these field campaigns, although the MACPEX campaign used an earlier version of a shattering probe that is slightly different compared with the ones used in later NASA campaigns. 2DS processing software also includes shattering removal algorithms (Lawson, 2011). For two research flights in ATTREX (RF03 and RF07), the FCDP probe did not provide measurements and

therefore the Hawkeye-CDP probe was used to provide the same size range (1 – 50 µm) of measurements.

Several additional steps were taken to derive ice microphysical properties from the key measurements mentioned above. For 2DS, CDP, FCDP, and Hawkeye-CDP probes, their measurements in the first bin were discarded to avoid possible uncertainties in that bin. A similar procedure of discarding small size particles in 2DS measurements has also been applied in a previous study by Mitchell et al. (2018). For the Fast-2DC probe, the first 3 bins were discarded to minimize uncertainties,

and the last 6 bins were discarded to reach a similar size range as the 2DS probe. After these procedures, the measurements of these probes were combined. That is, in the NSF campaigns, the CDP probe measurements were combined with the Fast-2DC probe measurements, providing a final size range of 2 – 3012.5 µm. In the NASA ATTREX, POSIDON and SEAC$^4$RS campaigns, 2DS measurements were restricted to 15 – 3005 µm and then combined with FCDP (or Hawkeye-CDP) measurements at 1 – 14.5 µm, which produced a combined size range of 1 – 3005 µm. Since NASA DC3 and MACPEX did

not have FCDP, only 2DS measurements were used for the size range of 15 – 3005 µm after discarding the first bin of 2DS. In summary, the compiled dataset of all NSF campaigns provided a final range of 2 – 3012.5 µm, while the compiled dataset of all NASA campaigns provided a final range of 1 – 3005 µm. The size range of the combined dataset for all NASA + NSF campaigns was 1 – 3012.5 µm. Combined NASA + NSF campaigns with the size range of 1 – 3012.5 µm were used for all tables and figures in the main manuscripts, including Tables 1 – 3, Figures 1 – 3, and 5 – 10, and all the analyses shown in the

supplemental material. The separate NSF and NASA campaigns were analyzed in Figure 4 and part of Figure 5 to contrast the differences between these campaigns.

Furthermore, IWC, Ni, and Di were calculated for the combined size range for each flight campaign. IWC was derived based on the mass-dimensional relationship following Brown and Francis (1995). For both NASA and NSF datasets, the in-cloud condition is defined when ice crystals have been detected in a 1-second measurement, that is, Ni > 0 for either Fast-2DC or

2DS measurements. The rest of the samples are defined as the clear-sky condition. For the cirrus temperature regime, 730 flight hours were obtained at temperatures ≤ -40 °C (i.e., 251 and 479 hours from NSF and NASA datasets, respectively), which include 161.6 hours of in-cloud conditions (i.e., 81.7 and 80.0 hours from NSF and NASA datasets, respectively). More information regarding the flight hours for each flight campaign in the cirrus temperature range, i.e., temperatures ≤ -40 ºC is shown in supplemental Table S1. The hours of measurements are separately shown for all-sky, clear-sky, and in-cloud

conditions, as well as cirrus under two types of environmental conditions.

## 2.2 Design of the Machine Learning (ML) Models

Machine learning models were developed to examine the influences of various factors on the occurrences of cirrus clouds and their microphysical properties. The key variables investigated include temperature, RHi, $w$, Na$_{500}$, and Na$_{100}$. In previous





studies such as Patnaude and Diao (2020) and Maciel et al. (2023), other methods were developed to individually examine the

thermodynamic, dynamic, and AIE on cirrus microphysical properties. For example, by using a "delta-delta" method that removes the temperature effects on cirrus microphysical properties, linear regressions can be applied to quantify the correlations between fluctuations of a certain environmental factor and the fluctuations of a cirrus microphysical property. However, one limitation of such analysis is the lack of comparisons of the effects of multiple factors. Thus, to achieve a direct comparison of effects of multiple factors, an ML approach was developed in this work.

The entire observation dataset was pre-portioned into two distinct and separate parts to "train" and "test" the ML models. The entire observation data of each research flight were first separated into 10 consecutive flight segments. Seven of the 10 flight segments were randomly selected to be used as the training data, while the remaining three flight segments were used as the testing data. Another method of separating training and testing data was also investigated, which randomly selected 70 % of the 1-Hz data of a research flight as training data and the rest (30 %) as testing data. These two data separation methods show

similar results. Only the former segment-based separation method is illustrated in the following sections. Compared with the latter separation method, the segment-based method avoids possible correlations between training and testing datasets when separated at 1-Hz resolution. Another step taken to pre-process the data was the utilization of a "listwise deletion" method for data filtering. This deletion method was applied if any second of the observational datasets contained either temperatures > -40 °C or if any key variable of that second showed "NAN", then all variables of that entire second were removed from the dataset.

Using a classification ensemble algorithm, we create a random forest model consisting of 100 individual and distinct decision trees. In addition, a "Random Undersampling Boosting" (RUSBoost) algorithm was implemented to account for any imbalances of samples among various categories in the dataset to keep any training biases to a minimum. For example, in this dataset, flight hours of each campaign were dominated by clear-sky conditions rather than in-cloud conditions. The RUSBoost algorithm accounts for the disproportionate sampling of in-cloud conditions and randomly boosts the under-sampled category.

Three experiments were designed for the ML models (hereafter referred to as Tests A, B, and C), which aimed to answer the following science questions respectively: (1) Which factor(s) are more important for the ML model to predict the occurrences of cirrus clouds? (2) Which factor(s) are more important for the ML model to predict the fluctuations of IWC inside cirrus clouds? (3) Which factor(s) are more important for the ML model to predict the distributions of IWC as a function of temperature, RHi, and $w$ inside cirrus? The details of the ML analysis are shown in Section 3.5.

## 3 Results

### 3.1 Distributions of RHi and $\sigma_w$ for Cirrus Clouds in Two Environmental Conditions

Influences of thermodynamic (i.e., temperature and RHi) and dynamical conditions ($w$) are investigated for various types of cirrus clouds (Figures 2 and 3). Cirrus clouds were categorized into two types of conditions, depending on the fluctuations of $w$ in the adjacent environment. That is, for one second of measurement, if the region of ± 30 seconds surrounding it experienced

updrafts and downdrafts exceeding ±1 m s$^{-1}$ (i.e., $w \leq -1$ m s$^{-1}$ or $\geq 1$ m s$^{-1}$), then this 1-second observation was defined as non-





quiescent conditions. Previous airborne observations of cirrus clouds around convective activity showed frequent occurrences of $w \leq$ -1 m s$^{-1}$ or $\geq$ 1 m s$^{-1}$ (e.g., D'Alessandro et al., 2017; Diao et al., 2017). In addition, the rest of the observations experiencing smaller updrafts and downdrafts within ±1 m s$^{-1}$ are defined as vertically quiescent conditions. The observations of cirrus clouds under non-quiescent and vertically quiescent conditions are 52 and 110 hours, respectively. Note that because

of the nature of Eulerian-view sampling of research aircraft, this separation of two types of cirrus differs from the previous study that used Lagrangian trajectories of $w$ from model simulations to separate cirrus origins, i.e., convective (liquid) cirrus versus in-situ cirrus (Krämer et al., 2016, 2020). Global maps and vertical profiles of cirrus cloud observations in vertically quiescent and non-quiescent conditions are depicted in supplemental Figure S1. In addition, clear-sky samples in two environmental conditions at temperatures $\leq$ -40 °C are shown in Figure S2. The vertical distributions of IWC, Ni, Di, and

water vapor volume mixing ratio under two environmental conditions are illustrated in Figure S3.

Distributions of 1-Hz observations of RHi as a function of temperature are examined for cirrus under two environmental conditions separately using the combined datasets of NASA and NSF campaigns (Figure 2). In addition, the RHi – T distributions for clear-sky conditions under two environmental conditions are shown in supplemental Figure S4. The six latitudinal regions were individually analysed, including the Northern Tropical regions (NT), Northern Midlatitudes (NM),

Northern Polar regions (NP), Southern Tropical regions (ST), Southern Midlatitudes (SM), and Southern Polar regions (SP). The in-cloud conditions show higher frequencies of RHi concentrated within ± 20 % around the ice saturation line. On the other hand, clear-sky conditions (Figure S4) indicate higher variabilities in RHi. Higher frequencies of RHi > 140 % are seen in the tropical regions in both in-cloud and clear-sky conditions, while for the midlatitude and polar regions, the RHi samples are seen below the homogeneous freezing line (such as below 140 %), indicating a possible dominant role of heterogeneous

freezing based on the available thermodynamic conditions. This result is consistent with the finding of Cziczo et al. (2013) and Patnaude et al. (2021a) for the extratropical regions. More occurrences of RHi exceeding the homogeneous freezing threshold (around 160 % to 190 %) are seen in the NT region at temperatures below -55°C, associated with large fluctuations of vertical velocity in Figure 3, indicating that this region is more likely to initiate homogeneous freezing compared with other regions. In addition, these higher RHi values in the NT are seen in cirrus clouds under both non-quiescent and vertically

quiescent conditions, indicating that homogeneous freezing in the tropics is not only restricted to conditions with stronger updrafts and downdrafts and plays an important role for the formation of both types of cirrus.

Similar to Figure 2, distributions of the standard deviations of $w$ (denoted as $\sigma_w$) are examined against various temperatures for both types of cirrus (Figure 3). The distributions of $\sigma_w$ for clear-sky conditions under non-quiescent and vertically quiescent conditions are shown in supplemental Figure S5. Here $\sigma_w$ are defined as the standard deviation of $w$ for the 1-Hz observations

calculated for every 10 km of aircraft observations. Most of the cirrus clouds in two conditions show $\sigma_w$ within 0.5 m s$^{-1}$. For the non-quiescent cirrus, the maximum $\sigma_w$ values range from 0.5 to 5 m s$^{-1}$ at various temperatures, which is a wider range compared with the vertically quiescent cirrus at 0.5 to 3 m s$^{-1}$. Comparing among different regions, the highest $\sigma_w$ values are seen in the NT and NM regions, where a few samples of $\sigma_w$ are seen to reach a maximum at 4 to 5 m s$^{-1}$.



### 3.2 Thermodynamic and Dynamical Controlling Factors on Cirrus Microphysical Properties

Three cirrus microphysical properties (IWC, Ni, and Di) are examined separately for NASA and NSF flight campaigns at various temperatures in Figure 4a–c and Figure 4d–f, respectively. Compared with the NSF campaigns which sampled the minimum temperature at -78.3 °C, the NASA ATTREX and POSIDON campaigns sampled temperatures as low as -88.2 °C. For both NASA and NSF campaigns, an increasing trend of average IWC with increasing temperatures is seen, which is consistent with previous observational studies of the IWC – T relationship (e.g., Diao et al., 2014a; Woods et al., 2018; Krämer
et al., 2020; Patnaude and Diao, 2020). In addition, a positive Di – T relationship is also seen, likely due to faster ice crystal growth under higher water vapor partial pressure and more sedimentation of larger ice crystals into lower altitudes with higher temperatures. Both NASA and NSF datasets show a nonlinear trend of Ni with increasing temperatures. The main difference between NASA and NSF datasets is that NASA dataset shows higher IWC and higher Ni by 0.5 order of magnitude, likely due to differences in cirrus microphysical properties at different geographical locations as previously discussed in Patnaude et
al. (2021a).

The relationships between the variability of cirrus ice microphysical properties and the variability of thermodynamic and dynamical conditions are further investigated in Figure 4 g – r. A "delta-delta" method is applied to various factors, similar to the method used in the study of Patnaude and Diao (2020), Patnaude et al. (2021a), and Maciel et al. (2023). Specifically, the delta value is calculated by subtracting the average value of a certain variable in each 1 °C temperature bin from every 1-
second datum based on the temperature bin that second belongs to. The calculation of the delta values removes the temperature effect. In addition, the average values of each 1 °C temperature bin are calculated for individual campaigns, therefore subtracting these campaign-specific average values reduces the impacts of geographical locations and different measurement platforms on these delta variables.

When examining the relationships of fluctuations of IWC, Ni, and Di (i.e., $dlog_{10}IWC$, $dlog_{10}Ni$, and $dlog_{10}Di$) with respect to
the fluctuations of temperature, RHi, and $w$ (i.e., dT, dRHi, and d$w$, respectively), the observed relationships are much more similar between the NASA and NSF datasets, which is reflected by the similar increasing or decreasing trends and similar ranges of delta values at various conditions between the two datasets. For example, both NASA and NSF datasets show a peak of $dlog_{10}IWC$ and $dlog_{10}Ni$ at dRHi slightly above 0 % (i.e., dRHi of 10 %–20 %). This result is consistent with that seen in Patnaude and Diao (2020), indicating that after ice nucleation, the continuous ice crystal growth and new ice particle formation
with sustained ice supersaturation will likely lead to the highest IWC and Ni. The decreasing trend of $dlog_{10}IWC$, $dlog_{10}Ni$, and $dlog_{10}Di$ with decreasing dRHi is also consistent with the previous studies of Diao et al. (2013, 2014b), which showed a decreasing trend of IWC, Ni, and Di with decreasing RHi during the sedimentation phase of cirrus cloud evolution.

As for the relationship with vertical velocity fluctuations, the maximum $dlog_{10}IWC$ and $dlog_{10}Ni$ are seen at the strongest updrafts and downdrafts, while the minimum $dlog_{10}IWC$ and $dlog_{10}Ni$ are seen associated with weak downdrafts (i.e., d$w$
around -0.25 to -0.75 m s$^{-1}$). This result indicates that large updrafts, which often are in close proximity to large downdrafts during turbulence and gravity waves (e.g., Diao et al. 2017), may provide sustained ice supersaturated conditions, and therefore





leading to the continuous formation of new ice particles. As for $dlog_{10}Di$ values, they reach maximum values when dRHi is around 20 % to 60 %, but remain relatively constant under various d$w$ values.

**3.3 Aerosol Indirect Effects on Cirrus Microphysical Properties**

AIEs on cirrus microphysical properties are investigated in Figure 5, which uses a delta-delta method similar to Figure 4. Three types of datasets are examined – NASA only (rows 1 and 4), NSF only (rows 2 and 5), and the combined NASA+NSF dataset (rows 3 and 6). The AIEs are separately examined for larger and smaller aerosols, i.e., $Na_{500}$ and $Na_{100}$ correspond to aerosol number concentrations when the particle diameter is greater than 500 nm and 100 nm, respectively. Understanding the correlations of aerosols with cirrus microphysical properties can give clues to the two main ice nucleation mechanisms.

Previously, aerosols larger than 500 nm have been used as a proxy for INPs when the direct measurements of INP are not available (DeMott et al., 2010). Note that due to the limitations of former INP measurement techniques, that study focused on temperatures higher than -30 °C instead of the cirrus cloud regime (i.e., $\leq$ -40 °C). Other studies using the particle analysis by laser mass spectrometry (PALMS) instrument showed that particles with diameters > 500 nm are dominated by dust particles and nonvolatile sea-salt for number and mass concentrations (Murphy et al., 2019; Froyd et al., 2019). Both dust (e.g., Hoose

and Möhler, 2012; Roesch et al., 2021) and sea salt (e.g., Patnaude et al., 2021b, 2024) have been previously reported to initiate heterogeneous freezing as INPs, which supports the speculation that $Na_{500}$ may be used as a proxy for INP number concentrations.

For the AIE of larger aerosols, a nearly linear positive correlation is seen in three cirrus microphysical properties (i.e., $dlog_{10}IWC$, $dlog_{10}Ni$, and $dlog_{10}Di$) in relation to $dlog_{10}Na_{500}$. The smaller aerosols show nonlinear correlations with cirrus

microphysical properties, as illustrated by the significant increases in $dlog_{10}IWC$ and $dlog_{10}Ni$ values when $dlog_{10}Na_{100}$ exceeds 1. That is, when $dlog_{10}Na_{100}$ values are significantly above (by a factor of 10) the average values of a 1-degree temperature bin, significant impacts on cirrus microphysical properties are seen. This feature indicates that there may be a sharp increase in ice nucleation through homogeneous freezing when much higher $Na_{100}$ values are seen. Because of the lack of direct measurements of aerosol chemical compositions in these campaigns, we cannot determine whether this nonlinearity is

associated with a change in aerosol composition in addition to the changes of their number concentrations. These main features of AIE from larger and smaller aerosols are consistently seen for either NASA, NSF campaigns separately, or the combined NASA + NSF campaigns. Therefore, for the following analyses, the combined NASA+NSF datasets (i.e., 1 – 3012.5 µm) are used in the quantitative analyses based on linear regressions (Figure 6) or ML models (Figures 7 – 10).

**3.4 Quantifications of Aerosol Indirect Effects based on Linear Regressions**

AIEs on cirrus microphysics are further quantified through linear regressions between the fluctuations of cirrus properties and the fluctuations in aerosol number concentrations in Figure 6 for the combined NASA+NSF dataset. The AIEs are individually quantified for different thermodynamic and dynamical conditions, including various ranges of temperatures from -40 to -70 °C, dRHi from below -10 % to above 10 %, and d$w$ from below -0.5 m s$^{-1}$ to above 0.5 m s$^{-1}$. Geometric means of $dlog_{10}IWC$,



dlog$_{10}$Ni, and dlog$_{10}$Di are calculated for each bin of dlog$_{10}$Na$_{500}$ or dlog$_{10}$Na$_{100}$. The full information of slopes, intercepts, and

their standard deviations for all linear regressions shown in Figure 6 is stored in supplemental Table S2.

Positive correlations are seen for various temperature, dRHi, and d$w$ ranges. In addition, for every range, larger positive slope values are seen in relation to dlog$_{10}$Na$_{500}$ compared with dlog$_{10}$Na$_{100}$, indicating stronger AIEs from the larger aerosols on three microphysical properties. In addition, when comparing among different ranges of dRHi and d$w$, the variabilities among the slope and intercept values for these different linear regressions with respect to larger aerosols (Figure 6 a5–a7, a9–a11) are

smaller than those seen with respect to smaller aerosols (Figure 6 b5–b7, b9–b11). These results suggest that with the availability of potential INPs (using larger aerosols as an indicator), ice nucleation is less dependent upon thermodynamic and dynamic factors such as the magnitudes of RHi and the strength of updrafts. On the other hand, for smaller aerosols, activating ice nucleation has higher requirements for the appropriate thermodynamic and dynamic conditions. For the AIE of smaller aerosols, such dependence upon thermodynamic and dynamic conditions are even stronger when relatively fewer aerosols are

available, as shown by the large separation between the geometric mean of cirrus properties at the lower values of dlog$_{10}$Na$_{100}$. That is, when dlog$_{10}$Na$_{100}$ < 0, the dlog$_{10}$IWC and dlog$_{10}$Ni values are 1 order of magnitude higher at higher dRHi (i.e., dRHi > 10 %) or at larger d$w$ (i.e., > 0.5 m s$^{-1}$) compared with those at lower dRHi (≤ 10 %) or lower d$w$ (< -0.5 m s$^{-1}$), respectively. The dlog$_{10}$Di values are also higher by a factor of 2 – 3 at these higher dRHi and d$w$ ranges. As dlog$_{10}$Na$_{100}$ increases, the cirrus properties converge to similar values, indicating that higher concentrations of smaller aerosols may also associate with

higher INP number concentrations, thereby lowering the requirements of the high RHi and $w$ thresholds.

## 3.5 Using Machine Learning (ML) Models to Quantify and Compare Thermodynamic and Dynamic Effects and Aerosol Indirect Effects on Cirrus Clouds

Three experiments are designed to quantify the contributions of various factors to cirrus cloud formation and the subsequent microphysical properties. ML models are designed to directly compare the contributions from temperature, RHi, $w$, Na$_{500}$, and

Na$_{100}$. Three ML tests in this section will be referred to as Tests A, B, and C. These three tests address the three scientific questions described in Section 3.2. That is, Test A examines the key factors contributing to the occurrences of cirrus clouds; Test B examines the key factors contributing to whether cirrus clouds are formed with higher and lower IWC values; and Test C examines the key factors contributing to the full range of magnitudes of IWC as a function of temperature, RHi, and $w$.

For this section, all the ML-based analysis uses the combined NASA+NSF dataset. When analyzing the effects of temperature

(T), RHi, and $w$ (e.g., Figures 7 and 8, top 4 rows), all 5 NASA and 7 NSF campaigns are included in the analysis. When analyzing the Na$_{500}$ and Na$_{100}$ variables, the NASA ATTREX and POSIDON campaigns are not included due to the lack of aerosol measurements (e.g., Figures 7 and 8, bottom 2 rows).

Test A trains the ML models to differentiate between clear-sky conditions and cirrus clouds. Because the prediction is for binary conditions (i.e., in-cloud versus out-of-cloud), Test A utilizes a binary ensemble classification algorithm for the ML

models. Results are analysed based on an accuracy scale of 0 – 100 %, to account for the percentage of 1-second samples being accurately predicted for its clear-sky or in-cloud condition. Various factors (e.g., T, RHi, $w$, Na$_{500}$, and Na$_{100}$), as well as a





combination of these factors, are used predictors in the ML models to examine which sets of variables provide more accurate predictions. Figure 7 shows 6 sets of predictors, including T, T+RHi, T+$w$, T+RHi+$w$, T+RHi+$w$+Na$_{500}$, and T+RHi+$w$+Na$_{100}$. Prediction results of more sets of predictors are shown in Table 2.

Results show that when using temperature solely as a predictor, 57.30 % accuracy is seen for all cirrus, while 57.69 % and 55.32 % accuracies are seen for two types of cirrus – vertically quiescent cirrus and non-quiescent cirrus, respectively. This indicates that when only providing temperature as the sole predictor, the chances of predicting cirrus formation is close to a random guess (i.e., 50 %). Besides the temperature predictor, other factors are added incrementally to examine the added values of these predictors. Among all of them, RHi is found to be most effective for enhancing prediction accuracy. The three

types of cirrus – all cirrus, vertically quiescent cirrus, and non-quiescent cirrus – show accuracies of 86.53 %, 86.41 %, and 87.16 %, respectively, when T+RHi predictors are used. Therefore, providing the additional information of RHi enhances the prediction from baseline T predictor by ~29 to 31 %. Comparatively, smaller increases of accuracies (by ~7 to 9%) are seen when T+$w$ are used, which show accuracies of 64.18 % and 66.39 % for all cirrus and vertically quiescent cirrus, respectively. Even lower accuracy (52.80 %) of predicting the occurrences of non-quiescent cirrus is seen by using the T+$w$ predictors

compared with using just the T predictor (55.32 %), likely caused by the pre-selection of dynamical conditions, which requires the existence of strong updrafts and downdrafts in the adjacent environments. That restriction already pre-selected the more favourable $w$ conditions and therefore making the $w$ factor less effective for enhancing the prediction accuracy any further.

When adding the predictors of aerosol information, the accuracies further increase by 2 % – 6 % compared with using T+RHi+$w$ (panel l), which are 91.97 %, 92.68 %, and 88.98 % when using T+RHi+$w$+Na$_{500}$, and 91.94 %, 92.62 % and 89.08

% when using T+RHi+$w$+Na$_{100}$ for three types of cirrus, respectively. Such increases of accuracy verify that AIEs do make a difference on the formation of cirrus clouds. Comparing between the larger and smaller aerosols, the differences in accuracy by using them as predictors are not very significant, which is within 0.1 %.

Table 2 shows more combinations of the five predictor variables, totalling to 23 sets of combinations. Using more predictors (i.e., T+RHi+$w$+Na$_{500}$ and T+RHi+$w$+Na$_{100}$) provides better results than using fewer predictors. All the tests that include RHi

as a predictor have accuracies exceeding 85 %, which show that RHi is consistently the most important factor among all five variables. Compared with RHi, $w$ plays a less important role in improving predictions of cirrus cloud occurrence regardless of being used as a single predictor or combined with other predictors. This result is likely caused by the fact that both water vapour concentrations and $w$ contribute to cooling rates that further control RHi magnitude, indicating that having the accurate representation of available water vapour concentrations is important besides the representation of dynamical conditions.

Test B is designed to examine what factors are more influential for the prediction of a cirrus cloud containing higher or lower IWC compared with the average conditions (Figure 8). Only in-cloud conditions are used for Test B. Here the predictors are calculated in terms of delta values, which are fluctuations relative to average values of every 1-degree temperature bin. Similar to Test A, a binary ensemble classification algorithm is used for Test B, predicting whether IWC is higher or lower than the average IWC in each 1-degree temperature bin (i.e., $dlog_{10}IWC > 0$ or $< 0$). Compared with the respective rows in Figure 7,

the accuracies for each set of predictors for predicting $dlog_{10}IWC > 0$ or $< 0$ are lower than the accuracies for predicting in-





cloud or out-of-cloud conditions. In fact, the accuracy of predicting the fluctuations of IWC does not exceed 77 % in any of the tests. This is likely due to the large variabilities of IWC in cirrus clouds, which can be several orders of magnitude different even within the same cirrus cloud layer. In addition, ice particle growth and formation of new ice particles all contribute to the variations in IWC, which require the understanding of the entire evolution of cirrus and the accumulative history of

environmental factors that the air parcel experienced.

When using dT as the sole predictor, the prediction has accuracies around 48 % to 50 %, which are closer to a random 50 % – 50 % guess. Adding dRHi to dT increases the accuracies to 64 % – 69 %, which indicates smaller increases of accuracies by adding dRHi as a predictor for IWC fluctuations compared with predicting cirrus occurrences in Figure 7. Adding $dw$ to dT increases the accuracies to 57 % to 59 %, indicating smaller contributions from $dw$ compared with dRHi for predicting the

fluctuations of IWC inside cirrus clouds. When adding aerosol information, the accuracies increase to 76.28 %, 76.49 %, and 76.11 %for the test of dT+dRHi+$dw$+dlog$_{10}$Na$_{500}$, and to 66.11 %, 65.26 %, and 66.83 % for dT+dRHi+ $dw$+dlog$_{10}$Na$_{100}$, for three cirrus types (i.e., all cirrus, vertically quiescent and non-quiescent), respectively. Compared between the larger and smaller aerosols, the added values of dlog$_{10}$Na$_{500}$ are 9 % to 12 % (by subtracting accuracies of panel l from panel o in Figure 7), while the added values of dlog$_{10}$Na$_{100}$ are slightly negative to nearly zero at -0.2 % to 1 % (by subtracting accuracies of

panel l from panel r in Figure 7). This result indicates that the larger aerosols play a more significant role in controlling the fluctuations of IWC compared with smaller aerosols. This result is consistent with the result shown in Figure 6, which shows higher positive slope values for correlations with dlog$_{10}$Na$_{500}$ (top 3 rows in Figure 6) compared with those for dlog$_{10}$Na$_{100}$ (bottom 3 rows in Figure 6). The stronger AIEs of larger aerosols on IWC inside cirrus are also consistent with previous studies using in-situ observations (e.g., Patnaude and Diao, 2020; Maciel et al., 2023). The added values of using larger aerosols as a

predictor in Test B (Figure 8) are higher than those seen in Test A (Figure 7), indicating that larger aerosols play a relatively more important role in controlling IWC fluctuations, possibly by modifying Ni and Di via ice nucleation, as well as by modifying the ambient RHi and $w$ via water vapor deposition and latent heat release, compared with a relatively weaker role for determining whether cirrus can be formed or not.

In addition to testing the effects of key factors at 1-Hz resolution as shown in Figure 8, we further examined the effects of

environmental factors on cirrus formation at coarser-scales from 10 km to 100 km in Table 3. Specifically, 50-s, 250-s, and 500-s averages of dT, dRHi, $dw$, dlog$_{10}$Na$_{500}$, dlog$_{10}$Na$_{100}$, and dlog$_{10}$IWC values are calculated surrounding each second, and these coarser-scale factors are used to predict whether the coarser-scale dlog$_{10}$IWC is above or below zero. This experiment addresses the question as to whether the IWC fluctuations are affected by larger-scale conditions, and what spatial scales are more impactful. For the effects of dRHi (using dT+dRHi as predictors), the accuracies of predicting the sign of dlog$_{10}$IWC for

vertically quiescent cirrus are 64.03 %, 70.33 %, 69.42 %, and 71.67 % for 1-s, 50-s, 250-s, and 500-s averaged observations, respectively, indicating the coarser-scale RHi conditions have larger impacts on the IWC fluctuations in vertically quiescent cirrus. This is likely because a higher RHi for a wider spatial scale can provide a favorable condition for ice crystal formation and growth for a larger cloud segment. For the effects of $dw$ (using dT+$dw$ as predictors) on vertically quiescent cirrus, the accuracies are 56.51 %, 57.55 %, 56.10 %, and 55.96 %, respectively, indicating that the effects of $w$ on IWC fluctuations is





more local and therefore it is more important to quantify the small-scale fluctuations in *w*. On the other hand, examining the

non-quiescent cirrus, even though the dT+dRHi prediction provides the highest accuracy of 80.56 % by using 250-s averaged

observations, the 500-s averaged observations provide the lowest accuracy of 66.32 % among all spatial scales, indicating a

sudden decrease in the impacts of RHi conditions beyond 50 km surrounding non-quiescent cirrus. For the d*w* on non-quiescent

cirrus, the accuracies show more variabilities, with only 43.87 % accuracy for 250-s averaged observations, indicating that

effects of d*w* on non-quiescent cirrus originate from a smaller surrounding environment.

For the analysis of AIEs, effects of $Na_{500}$ on vertically quiescent cirrus are larger for the more adjacent environments (i.e., 1-

Hz observations) than the coarser-scale environments. On the other hand, the effects of $Na_{100}$ on vertically quiescent cirrus are

larger for the coarser-scale environments (i.e., 500-s scale). This feature is likely due to larger aerosols (potentially serving as

INPs) significantly increasing the likelihood of forming ice particles at lower thresholds of RHi and *w*, while smaller aerosols

still require more restrictive thermodynamic and dynamic conditions to be satisfied. Therefore, a higher average $Na_{100}$ value

at a coarser scale are more likely to overlap with favorable RHi and *w* conditions. For non-quiescent cirrus, the effects of

aerosols show similar nonlinearity as seen in the effects of dRHi and d*w*. This is likely caused by the large spatial

heterogeneities of both environmental conditions and cloud microphysical properties surrounding the non-quiescent cirrus.

Test C examines the ability of the ML models to predict the distributions of IWC as a function of temperature, RHi, and *w*,

shown in Figures 9 and 10. The distributions based on real in-situ observations (Figure 9 a – c) show four main features: (1)

an increasing trend of IWC with increasing temperatures, (2) peak IWC values under small ice supersaturation (i.e., RHi of

110 %), (3) higher IWC at stronger updrafts and downdrafts, and (4) higher geometric mean IWC values in the non-quiescent

cirrus than the vertically quiescent cirrus by 1 order of magnitude. The higher IWC seen in non-quiescent cirrus is consistent

with the finding of Krämer et al. (2016) in their Figure 13. Three sets of predictions are evaluated, including T, T+RHi+*w*, and

T+RHi+*w*+$Na_{500}$+$Na_{100}$. All the tests can capture the first feature (positive correlations between IWC and T), but the test using

only T as a predictor cannot capture the trend with respect to RHi and *w*, nor can it show the different IWC between two types

of cirrus. Using T+RHi+*w* predictors can already capture the main differences in IWC between two types of cirrus. Adding

aerosols as predictors shows more similar results to observations at -75 to -65 °C and -50 to -40°C compared with only using

T+RHi+*w*, which illustrates aerosol indirect effects in addition to thermodynamic and dynamic effects.

Figure 10 a – f shows the comparisons of predicted IWC versus observed IWC, color coded by the average T, RHi, and *w*

within each bin. In addition, Figure 10 g – l compares the probability density functions (PDFs) of T, RHi and *w* between the

scenarios when ML models underestimate or overestimate IWC values. When RHi is not included as a predictor, the predicted

IWC values are underestimated at higher RHi values (i.e., orange and red bins below the 1:1 line in panel b) and overestimated

at lower RHi values (i.e., blue bins above 1:1 line). In addition, when only using T as the predictor in panel h, the ML

predictions overestimating IWC (red line) show higher frequencies of subsaturated conditions and lower frequencies of ice

supersaturated conditions, compared with the ML predictions that underestimate IWC. Similarly, when w is excluded from the

prediction, the higher IWC values associated with strong updrafts are underestimated (i.e., red bins under 1:1 line in panel c).

The PDFs of *w* also show that the underestimated IWC samples have higher frequencies of strong updrafts and downdrafts





when $w$ is not used as a predictor in panel i. The differences in PDFs of RHi and $w$ between overestimated and underestimated
IWC samples are much smaller when all three predictors are used (i.e., T+RHi+$w$) in panels k and l. These analyses
demonstrate the importance of accurately representing the RHi and $w$ distributions in model simulations when simulating the
magnitudes of IWC in cirrus clouds.

## 4 Conclusions and Implications

In this study, near global-scale datasets were developed for in-situ observations of cirrus microphysical properties and their
surrounding environmental conditions. Individual roles of several key factors (i.e., temperature, RHi, $w$, $Na_{500}$, and $Na_{100}$)
affecting the distributions of cirrus microphysical properties were investigated. The datasets cover a wide range of latitudes,
providing observations in six latitudinal bands ranging from the polar regions to the midlatitudes and the tropics.

Several approaches were developed to quantify these individual effects, including using a "delta-delta" method to examine the
correlations between the fluctuations of environmental conditions and the fluctuations of cirrus properties, using linear
regressions to quantify the aerosol indirect effects of larger and smaller aerosols, and using random forest ML models to
address the effectiveness of adding different variables as predictors for predicting the occurrences of cirrus and the subsequent
IWC fluctuations and magnitudes. These methods have been shown to be critical for quantifying the role of different factors.
For instance, the effects of RHi and $w$ on IWC, Ni and Di were examined by removing the temperature effects on cirrus
properties in Figure 5. The five NASA and seven NSF campaigns show similar trends when the fluctuations of IWC, Ni and
Di were examined, including the peak of $dlog_{10}IWC$ and $dlog_{10}Ni$ seen at 10 % dRHi, and the peak of $dlog_{10}IWC$ and $dlog_{10}Ni$
seen at stronger updrafts and downdrafts conditions. The calculation of delta values enables the combination of NASA and
NSF datasets for linear regression analysis of AIEs (Figure 6). The average background conditions were subtracted from the
delta values, removing the variabilities introduced by various instruments and geographical locations.

The ML models were designed to directly compare the effects of multiple factors (Figures 7 – 10 and Tables 2 and 3). Among
all factors, RHi is the most important factor for predicting the occurrences of cirrus clouds, although its relative contributions
to the fluctuations and magnitudes of IWC are smaller compared with its dominant role for predicting cirrus occurrences.
Comparing between non-quiescent and vertically quiescent cirrus, the non-quiescent cirrus clouds show 1 order of magnitude
higher IWC than vertically quiescent cirrus, which can be captured if the predictors of T+RHi+$w$ are used.

For the AIEs, both larger and smaller aerosol concentrations ($Na_{500}$ and $Na_{100}$) show positive correlations with the delta values
of IWC, Ni, and Di when the combined NASA+NSF datasets were examined. However, larger aerosols produce stronger AIE
(i.e., steeper slopes) than smaller aerosols shown by the slopes of linear regressions (Figure 6). In addition, near-linear
correlations with positive slopes are seen between fluctuations of IWC, Ni, and Di relative to fluctuations of larger aerosols,
while the correlations with smaller aerosols are nonlinear. The increasing trend of $dlog_{10}IWC$, $dlog_{10}Ni$, and $dlog_{10}Di$ become
more visible when the number concentrations of smaller aerosols are 10 times larger than their background conditions (i.e.,
$dlog_{10}Na_{100} > 1$). This is likely because larger aerosols are more likely to freeze via the heterogeneous nucleation, while the





smaller aerosols are more likely to freeze via homogeneous freezing. For AIE of large aerosols based on ML analysis, their relative contributions for the cirrus occurrence are relatively small compared with those from RHi and $w$ (Figure 7), but their relative contributions for the IWC magnitudes are comparable to those from dRHi and d$w$ (Figure 8). For the AIE of small aerosols, they do not significantly contribute to IWC fluctuations (Figure 8), but contribute to cirrus occurrences on a similar

magnitude as the larger aerosols do (Figure 7).

When examining the impacts of using predictors at different spatial scales, the larger aerosols are more effective for predicting IWC fluctuations within a close proximity to the 1-Hz in-cloud samples, while the smaller aerosol concentrations at coarser scales are more effective for predicting IWC fluctuations. These results indicate that the existence of large aerosols, which may serve as INPs, is more likely to be the sufficient condition for ice nucleation since it lowers the requirements of other

conditions, such as by lowering the RHi thresholds. Therefore, $Na_{500}$ is more likely to contribute to the prediction of ice microphysics properties at the same location. On the other hand, the existence of small aerosols is more likely a necessary, but not sufficient, condition. That is, higher $Na_{100}$ values in a larger spatial domain may provide a background condition to support ice nucleation but other conditions, such as a relatively higher RHi threshold, still need to be satisfied.

The compiled datasets show a significantly lower number of in-situ measurements of cirrus clouds in the polar regions (i.e.,

both NP and SP) compared with the other latitudinal regions in the midlatitudes and tropics (Figure 2). In addition, most of the observations of cirrus clouds in the midlatitudes (except for MACPEX) obtained from these former field campaigns were targets of opportunities that were captured en route instead of being the main scientific objectives of those campaigns. Thus, more airborne field campaigns in the mid- and high latitudes are needed to understand the key environmental factors controlling cirrus formation and evolution by specifically targeting the cirrus cloud system. More comparative studies among cirrus clouds

formed under various synoptic dynamical conditions (i.e., convective, orographic, and in-situ cirrus) are also still warranted in order to examine the controlling factors on different types of cirrus.

Quantifying the relative role of various factors has implications for improving the simulations of cirrus clouds in GCMs. For example, capturing the exact timing and location of the concentrations of larger aerosols is more important than capturing such information for small aerosols, especially for predicting variabilities of IWC (Table 3). In addition, the formation of both

vertically quiescent and non-quiescent cirrus clouds requires accurate presentations of RHi at various horizontal scales from 0.2 – 100 km, which presents a challenge to sub-grid parameterizations in GCMs. Overall, this study provided two main types of metrics to quantify the contributions from multiple factors on cirrus microphysical properties, i.e., linear regressions and ML predictions. These datasets and metrics developed in this study can be applied to evaluate GCM simulations and satellite-based observations for cirrus microphysical properties and AIEs on cirrus clouds.

**Data availability**

Observations from the seven NSF flight campaigns are accessible at https://data.eol.ucar.edu/. Observations from the five NASA flight campaigns are accessible at https://www-air.larc.nasa.gov/missions.html.



**Author contributions**

D. Ngo and M. Diao contributed to the development of the ideas, conducted quality control to aircraft-based observations,
conducted data analysis, and wrote the manuscript. R. Patnaude contributed to the quality control of in-situ observations. S.
Woods contributed to the field maintenance, calibration, and final data processing for the 2DS probe. G. Diskin supported the
field measurements, calibration and final data submission for the DLH hydrometer.

**Competing interests**

The authors declare that they have no conflict of interest.

**Acknowledgments**

M. Diao, D. Ngo, and R. Patnaude acknowledge funding from NASA ROSES-2020 80NSSC21K1457, NSF AGS #1642291,
and NSF OPP #1744965 grants. D. Ngo and R. Patnaude also acknowledge support from the San Jose State University Walker
Fellowship.

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



**Table 1.** Descriptions of 5 NASA and 7 NSF campaigns used in this work, including their names, acronyms, times, locations and key instruments. Cirrus cloud observations including in-cloud flight hours ≤ -40°C and ranges of temperatures, altitudes and pressures are also provided.

| Field Campaign | Full Name | Time | Spatial Extent | Cirrus obs hours | Cirrus Sample Range (min / max) | Key Instruments |
|---|---|---|---|---|---|---|
| NSF HIPPO* | HIAPER Pole-to-pole Observations | Oct-Nov, 2009 Mar-Apr, 2010 Jun-July, 2011 Aug-Sep, 2011 | 67°S – 87°N, 128°E – 90°W | 6.29 | -77.2 – -40 °C 4.5 – 14.9 km 133 – 531 hPa | Fast-2DC, CDP, Rosemount, VCSEL, UHSAS |
| NSF START08 | Stratosphere-Troposphere Analyses of Regional Transport | Apr-Jun, 2008 | 26°N – 63°N, 117°W – 86°W | 2.28 | -67.7 – -40 °C 6.1 – 14.9 km 133 – 447 hPa | Fast-2DC, CDP, Rosemount, VCSEL, UHSAS |
| NASA SEAC⁴RS | Studies of Emissions and Atmospheric Composition, Clouds and Climate Coupling by Regional Surveys | Aug-Sept, 2013 | 19°N – 50°N, 80°W – 120°W | 4.71 | -59.5 – -40 °C 9.8 – 13.2 km 179 – 290 hPa | 2DS, FCDP, MMS, DLH, UHSAS |
| NSF DC3 | Deep Convective Clouds and Chemistry Project | May-Jun, 2012 | 25°N – 43°N, 106°W – 79°W | 22.89 | -65.9 – -40 °C 9 – 14.4 km 147 – 322 hPa | Fast-2DC, CDP, Rosemount, VCSEL, UHSAS |
| NASA DC3 | Deep Convective Clouds and Chemistry Project | May-Jun, 2012 | 30°N – 42°N, 117°W – 106°W | 14.45 | -63.5 – -40 °C 9.2 – 12.2 km 186 – 298 hPa | 2DS, MMS, DLH, UHSAS |
| NASA MACPEX | Mid-latitude Airborne Cirrus Properties EXperiment | Mar-Apr, 2011 | 26°N – 41°N, 104°W – 84°W | 13.00 | -77.3 – -40 °C 8.2 – 17.8 km 77 – 347 hPa | 2DS, MMS, HWV, FCAS |
| NSF CONTRAST | CONvective TRansport of Active Species in the Tropics | Jan-Feb, 2014 | 20°S – 40°N, 132°E – 105°W | 22.80 | -78.3 – -40 °C 8.6 – 15.3 km 127 – 332 hPa | Fast-2DC, CDP, Rosemount, VCSEL, UHSAS |
| NASA ATTREX-2014 | Airborne Tropical TRopopause EXperiment | Jan-Feb, 2014 | 12°S – 36°N, 134°E – 117°W | 31.97 | -88.2 – -40 °C 8.8 – 18.8 km 68 – 331 hPa | Hawkeye-2DS, FCDP, Hawkeye-CDP, MMS, DLH |
| NSF PREDICT | PRE-Depression Investigation of Cloud systems in the Tropics | Aug-Sep, 2010 | 10°N – 29°N, 87°W – 38°W | 17.33 | -71.4 – -40 °C 10.3 – 14.8 km 140 – 273 hPa | Fast-2DC, CDP, Rosemount, VCSEL, UHSAS |
| NASA POSIDON | Pacific Oxidants, Sulfur, Ice, Dehydration, and cONvection | Oct, 2016 | 1°S – 15°N, 131°E – 161°E | 12.65 | -87.9 – -40 °C 10.4 – 19.4 km 63 – 253 hPa | 2DS, FCDP, MMS, DLH |
| NSF TORERO | Tropical Ocean tRoposphere Exchange of Reactive halogen species and Oxygenated voc | Jan-Feb, 2012 | 42°S – 14°N, 105°W – 70°W | 1.89 | -75 – -40 °C 8.3 – 15.3 km 124 – 345 hPa | Fast-2DC, CDP, Rosemount, VCSEL, UHSAS |
| NSF ORCAS | The O₂/N₂ Ratio and CO₂ Airborne Southern Ocean Study | Jan-Mar, 2016 | 75°S – 18°S, 91°W – 51°W | 1.04 | -68.9 – -40 °C 6.3 – 13 km 176 – 433 hPa | Fast-2DC, CDP, Rosemount, VCSEL, UHSAS |

* Only used deployments #2 to #5.



**Table 2.** Summary of results for Test A, namely predicting the occurrences of cirrus clouds. Accuracies of the predictions are shown for all cirrus, vertically quiescent, and non-quiescent cirrus in columns 1 – 3, respectively.

| *Predictors* | *Accuracy (%)* *All cirrus* | *Accuracy (%)* *Vertically quiescent cirrus* | *Accuracy (%)* *Non-quiescent cirrus* |
|---|---|---|---|
| *Using T, RHi, and w as predictors* | | | |
| $T$ | 57.30 | 57.69 | 55.32 |
| $RHi$ | 85.30 | 85.01 | 86.79 |
| $w$ | 68.04 | 71.39 | 50.79 |
| $T + RHi$ | 86.53 | 86.41 | 87.16 |
| $T + w$ | 64.18 | 66.39 | 52.80 |
| $RHi + w$ | 85.41 | 85.16 | 86.73 |
| $T + RHi + w$ | 86.58 | 86.46 | 87.20 |
| *Using T, RHi, w, and $Na_{500}$ as predictors* | | | |
| $Na_{500}$ | 84.20 | 88.84 | 64.69 |
| $T + Na_{500}$ | 73.28 | 76.34 | 60.40 |
| $T + RHi + Na_{500}$ | 91.91 | 92.57 | 89.11 |
| $T + w + Na_{500}$ | 77.76 | 82.82 | 56.51 |
| $RHi + Na_{500}$ | 91.34 | 91.89 | 89.05 |
| $RHi + w + Na_{500}$ | 91.62 | 92.22 | 89.09 |
| $w + Na_{500}$ | 76.18 | 81.43 | 54.12 |
| $T + RHi + w + Na_{500}$ | 91.97 | 92.68 | 88.98 |
| *Using T, RHi, w, and $Na_{100}$ as predictors* | | | |
| $Na_{100}$ | 68.96 | 70.30 | 63.32 |
| $T + Na_{100}$ | 68.62 | 69.86 | 63.38 |
| $T + RHi + Na_{100}$ | 91.58 | 92.17 | 89.13 |
| $T + w + Na_{100}$ | 74.43 | 77.29 | 62.43 |
| $RHi + Na_{100}$ | 91.46 | 92.05 | 89.02 |
| $RHi + w + Na_{100}$ | 91.63 | 92.27 | 88.97 |
| $w + Na_{100}$ | 70.73 | 73.78 | 57.90 |
| $T + RHi + w + Na_{100}$ | 91.94 | 92.62 | 89.08 |





**Table 3.** Summary of results for Test B, namely predicting whether IWC inside cirrus is higher or lower than the average IWC
conditions. Similar to Table 2, accuracies of the predictions are shown for all cirrus, vertically quiescent, and non-quiescent
cirrus in columns 1 – 3, respectively. Effects of multiple factors are analyzed at different spatial scales, i.e., 1-s, 50-s, 250-s,
and 500-s averaged conditions.

| Predictors | Accuracy (%) All cirrus | Accuracy (%) Vertically quiescent cirrus | Accuracy (%) Non-quiescent cirrus |
|---|---|---|---|
| *1-Hz observations* | | | |
| $dT$ | 48.84 | 49.45 | 47.81 |
| $dT + dRHi$ | 65.76 | 64.03 | 68.67 |
| $dT + dw$ | 57.30 | 56.51 | 58.62 |
| $dT + dRHi + dw$ | 65.15 | 64.02 | 67.04 |
| $dT + dRHi + dw + dlog_{10}Na_{500}$ | 76.28 | 76.49 | 76.11 |
| $dT + dRHi + dw + dlog_{10}Na_{100}$ | 66.11 | 65.26 | 66.83 |
| *50-s averaged observations* | | | |
| $dT$ | 49.36 | 49.49 | 44.98 |
| $dT + dRHi$ | 70.33 | 70.33 | 70.34 |
| $dT + dw$ | 57.21 | 57.55 | 46.23 |
| $dT + dRHi + dw$ | 70.68 | 70.62 | 72.78 |
| $dT + dRHi + dw + dlog_{10}Na_{500}$ | 71.58 | 71.50 | 74.07 |
| $dT + dRHi + dw + dlog_{10}Na_{100}$ | 70.78 | 70.74 | 72.39 |
| *250-s averaged observations* | | | |
| $dT$ | 51.74 | 51.71 | 55.04 |
| $dT + dRHi$ | 69.50 | 69.42 | 80.56 |
| $dT + dw$ | 56.02 | 56.10 | 43.87 |
| $dT + dRHi + dw$ | 69.99 | 69.95 | 75.81 |
| $dT + dRHi + dw + dlog_{10}Na_{500}$ | 69.89 | 69.78 | 85.53 |
| $dT + dRHi + dw + dlog_{10}Na_{100}$ | 69.64 | 69.60 | 74.89 |
| *500-s averaged observations* | | | |
| $dT$ | 50.70 | 50.71 | 49.61 |
| $dT + dRHi$ | 71.65 | 71.67 | 66.32 |
| $dT + dw$ | 56.01 | 55.96 | 68.62 |
| $dT + dRHi + dw$ | 72.17 | 72.18 | 67.29 |
| $dT + dRHi + dw + dlog_{10}Na_{500}$ | 72.32 | 72.38 | 56.00 |
| $dT + dRHi + dw + dlog_{10}Na_{100}$ | 71.99 | 72.05 | 57.39 |



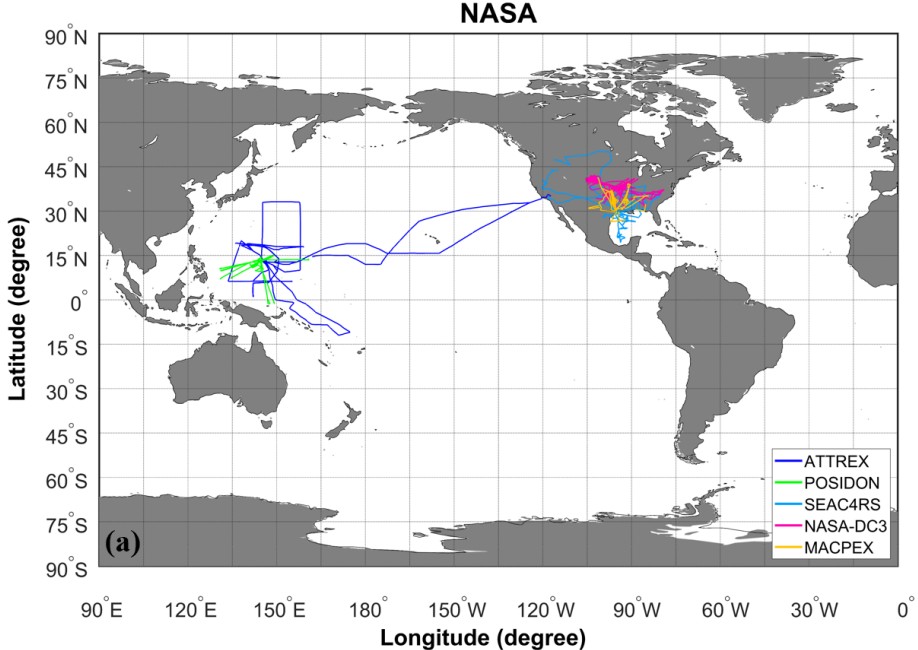

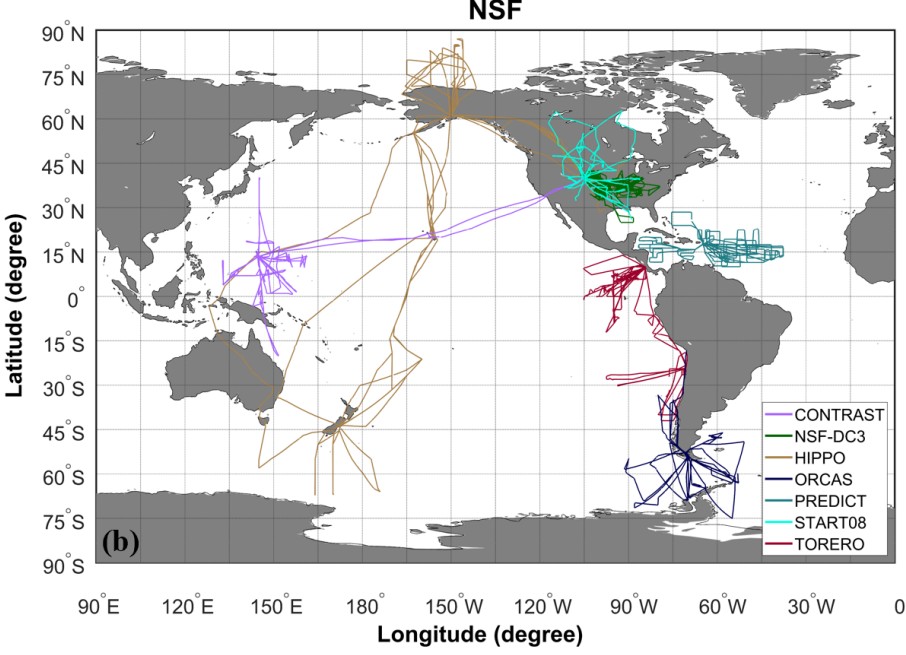

**Figure 1.** Global maps of research aircraft flight tracks from (a) five NASA campaigns and (b) seven NSF flight campaigns used in this observational study. The entire flight tracks at all temperatures as shown.





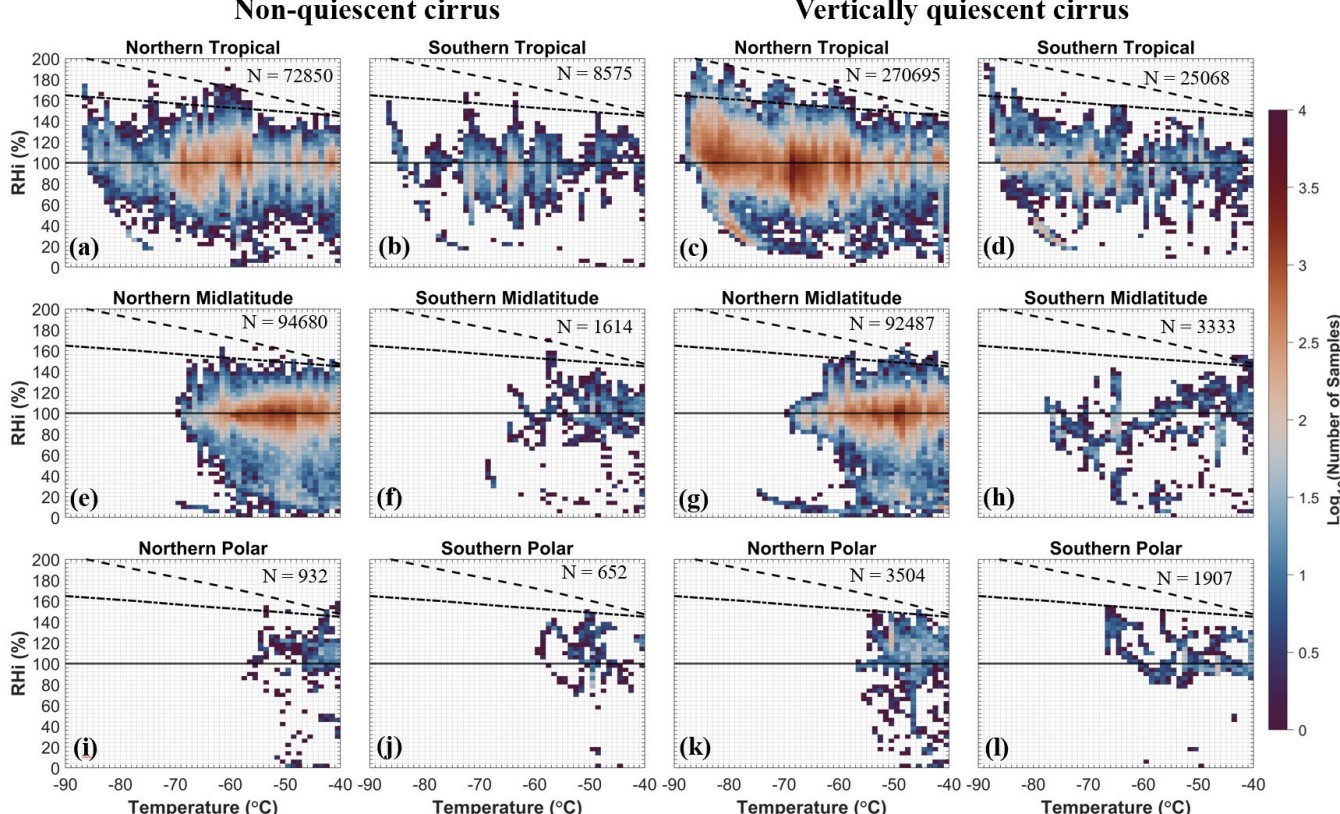

**Figure 2.** Distributions of RHi at various temperatures in 6 latitudinal bands using the combined NASA and NSF dataset, separated by non-quiescent cirrus (two left columns) and vertically quiescent cirrus (two right columns). Solid black line indicates ice saturation. Dashed black line denotes the liquid saturation threshold. Dash-dotted line represents the homogeneous freezing line based on Koop et al. (2000). Color bars denote logarithmic-scale number of samples.



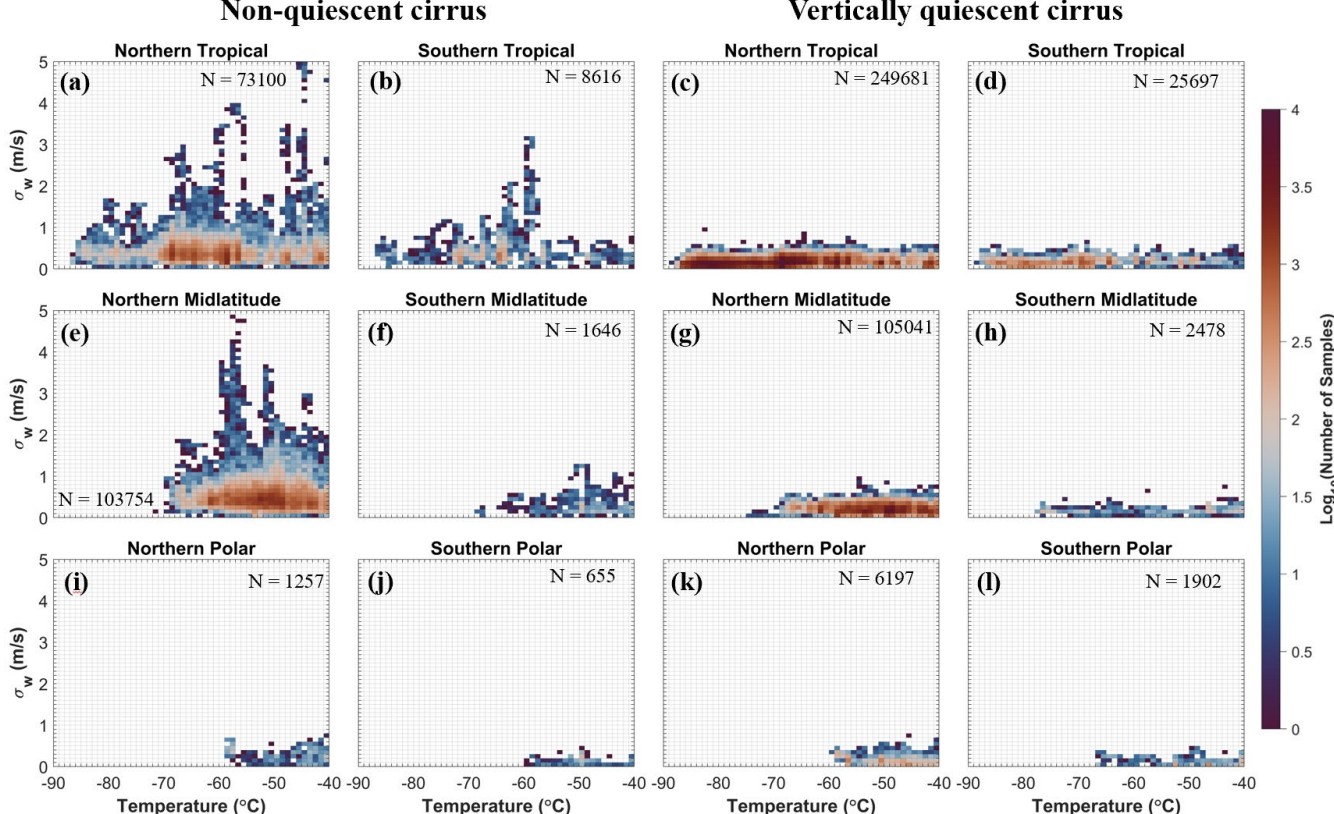

**Figure 3.** Distributions of standard deviations of vertical velocity ($\sigma_w$ calculated for 10 km spatial scales) at various temperatures, separated by non-quiescent cirrus (two left columns) and vertically quiescent cirrus (two right columns).





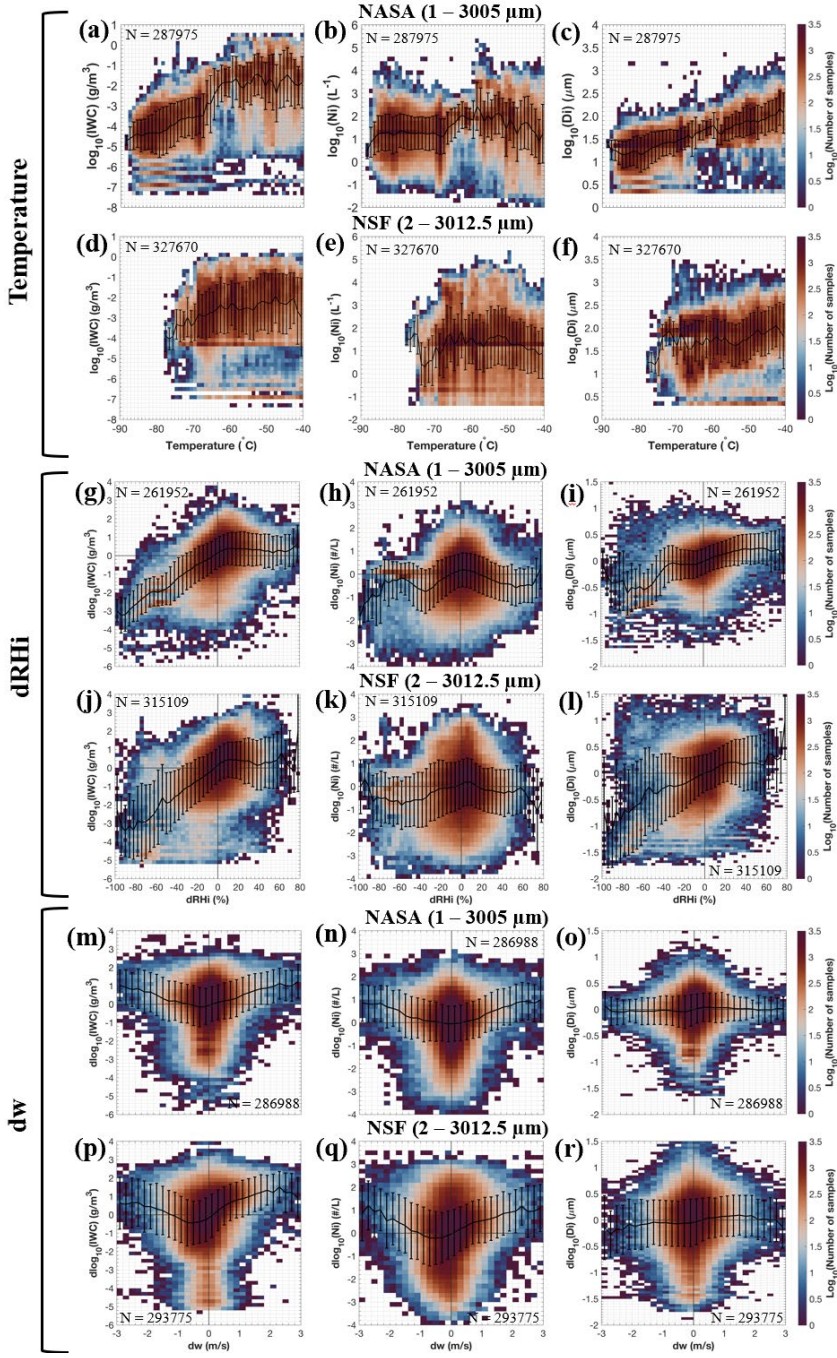

**Figure 4.** (a-f) Distributions of IWC, Ni, and Di as a function of temperature. Relationships between the (g-l) fluctuations of RHi (calculated as dRHi) and (m-r) fluctuations of $w$ (calculated as d$w$) with respect to the fluctuations of ice microphysical properties. Rows 1, 3, 5 are based on NASA campaigns and rows 2, 4, 6 are based on NSF campaigns. Black lines and vertical bars denote the geometric means and standard deviations, respectively.





**Figure 5.** Similar to Figure 4, but for relationships of fluctuations of cirrus properties (i.e., $dlog_{10}IWC$, $dlog_{10}Ni$, and $dlog_{10}Di$) with respect to $dlog_{10}(Na_{500})$ in top 3 rows and $dlog_{10}(Na_{100})$ in bottom 3 rows. Rows 1 and 4 are based on NASA campaigns, rows 2 and 5 are NSF campaigns, and rows 3 and 6 are the combined NASA+NSF campaigns.



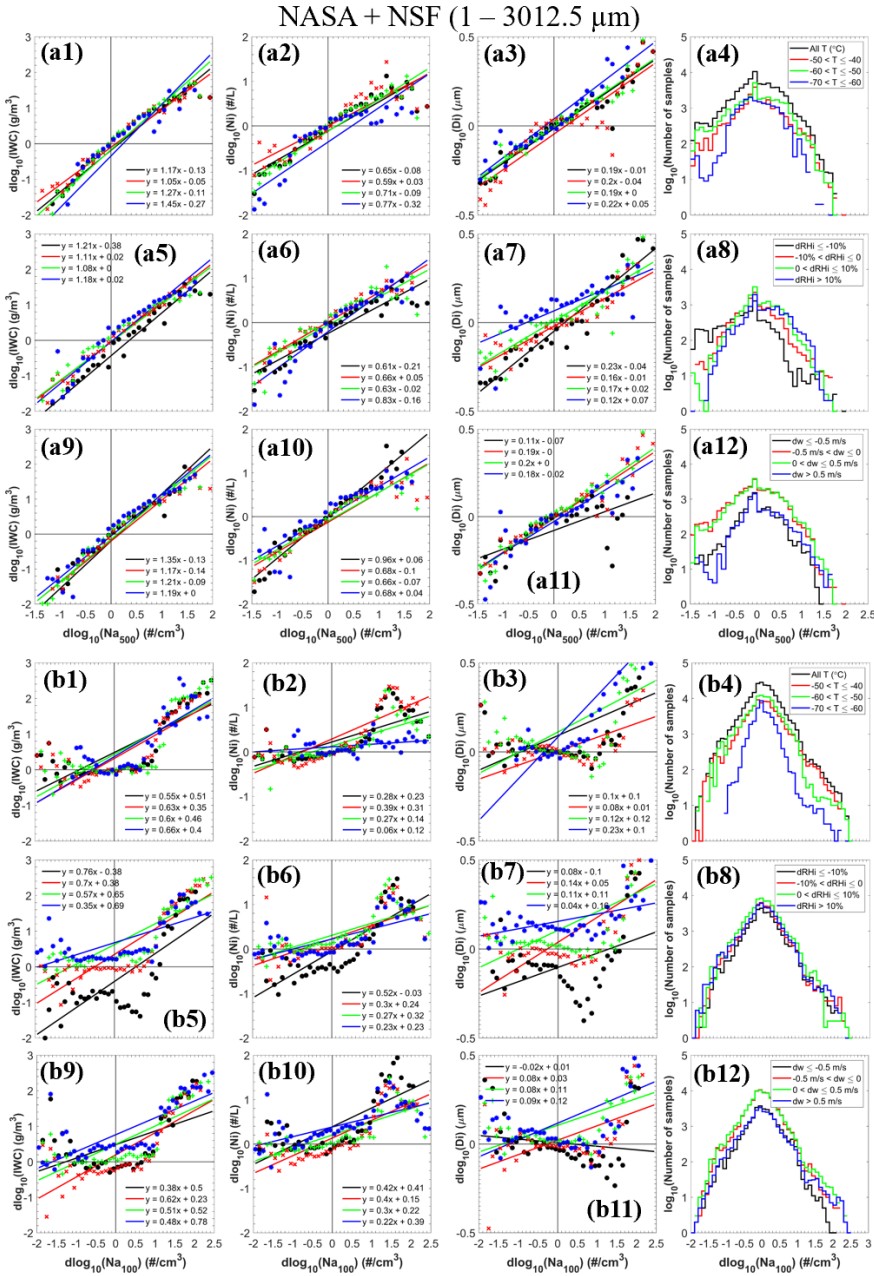

**Figure 6.** Linear regressions quantifying the correlations of $dlog_{10}IWC$, $dlog_{10}Ni$, and $dlog_{10}Di$ with respect to $dlog_{10}(Na_{500})$ in top 3 rows and $dlog_{10}(Na_{500})$ in bottom 3 rows. The analyses in Figures 6 – 10 use the combined NASA+NSF datasets (1 – 3012.5 μm). Aerosol indirect effects are examined for various ranges of temperature, dRHi, and d$w$. Colored dots represent geometric means of ice microphysical properties in each Na bin. Slope and intercept values are shown in the legend. The last column represents the distributions of the number of samples.



**Figure 7.** Prediction accuracies (in %) of Test A, namely using ML models to predict the binary condition of in-cloud or out-of-cloud for temperatures ≤ -40 °C. Columns 1 and 2 show the accuracies for predicting observed in-cloud and observed clear-sky conditions, respectively. Red and green indicate correct and false predictions, respectively. Column 3 shows the predication of three types of cirrus – all cirrus, vertically quiescent (VQ), and non-quiescent (NQ) cirrus. The set of predictors used in each test is labelled on the right-hand side of each row. ML predictions using T, RHi, and *w* are based on all 12 campaigns, while ATTREX and POSIDON are not included in the bottom 2 rows due to the lack of aerosol measurements.





**Figure 8.** Similar to Figure 7 but predicting whether $dlog_{10}IWC$ is positive (+) or negative (-) for in-cloud conditions. $dlog_{10}IWC$ is calculated relative to the geometric mean of IWC in each 1-degree temperature bin inside cirrus clouds. 1-Hz observations are used in this analysis compared with coarser scales used in Table 2. Columns 1 and 2 represent the scenarios when the real observations show $dlog_{10}IWC > 0$ and $< 0$, respectively. Column 3 shows the overall accuracies for predicting the sign of $dlog_{10}IWC$ in three types of cirrus.




**Figure 9.** Distributions of $\log_{10}$IWC in relation to temperature, RHi, and *w* in columns 1 – 3, respectively. Various sets of predictors are used in different rows. The solid horizontal lines and the vertical bars represent the geometric means and standard deviations of (a-c) observed and (d-l) predicted $\log_{10}$IWC. Red and blue represent results for non-quiescent and vertically quiescent cirrus, respectively.







**Figure 10**. (a-f) Distributions of predicted versus observed $\log_{10}$IWC colored coded by the average temperature, RHi, and $w$ in each bin for columns $1-3$, respectively. (g-l) PDFs of T, RHi, and $w$, separated by when IWC is underestimated or overestimated by the ML model. Rows 1 and 3 are predicted by T only; Rows 2 and 4 are predicted by T+RHi+$w$.