# Peer review of "Aerosol Indirect Effects on Cirrus Clouds Based on Global-Scale Airborne Observations and Machine Learning Models"

_EGUsphere, 2024_

## Referee Comment (RC1)

Ngo review

The authors evaluate relationships between cirrus microphysical properties and other relevant parameters using in situ observations from twelve field campaigns globally. The study focuses on comparing observations of microphysical properties with temperature, vertical air motion and aerosol concentrations using commonly deployed instrumentation. They perform multiple analyses - the two primary methodologies consist of comparisons of detrended datasets and another using a machine learning algorithm to determine how sufficiently the environmental parameters influence both the occurrence and properties of cirrus clouds.

The study addresses pressing questions concerning cirrus clouds, and figures throughout the manuscript were clear and well organized. However, I have a few major concerns which I discuss below. I also urge the authors to diligently proofread the manuscript, as many statements should be rewritten to improve clarity.

Major Concern #1:

It is common practice not to use aerosol measurements from spectrometers or CCN counters in cloud, but rather use measurements taken in clear-sky regions in the vicinity of the clouds of which they are compared to. This commonly involves obtaining measurements above or below cloud. I strongly suspect that Naer500 is sampling small ice particles, whether related to shattering or not, which is why such strong linear trends are observed in Figure 6 regardless of temperature, w, or RHi range.

It is crucial to show that measurements are not biased when using in-cloud measurements. Unfortunately, the large set of observations may make this a tedious task, but it is vital to assure these are accurate measurements as much of this manuscript evaluates aerosol-cloud interactions. I would recommend presenting multiple case studies from each campaign showing Naer500 in the vicinity of and within cloud.

Major Concern #2:

Perhaps not a major concern, but the in-cloud threshold is very low. Especially for OAP measurements that use particles down to diameters of 15 microns, this can result in in-cloud measurements below $10^{-7}$ g m$^{-3}$, which is an extremely low and questionable threshold considering the relevance in columnar radiative fluxes. It would be prudent to test findings using a higher threshold, and at least provide sources as evidence for what in-cloud thresholds should be considered for thin cirrus which significantly impact radiative fluxes. I also mention this here as a major concern since I wouldn't expect the aerosol measurements to be biased at extremely low cloud thresholds, which should be taken into consideration when evaluating potential biases.

Major Concern #3:

I worry the machine learning methodology is not sound. It seems the authors are using it to discern which environmental parameters cirrus cloud occurrence frequencies and microphysical properties are most sensitive to. However, they only use a select few combinations of predictor variables when evaluating the success rate of the model to simulate the observations for Test B and C. Due to the "black box" nature of the ML method, we need to test all possible combinations of the predictor variables and weight their success rates accordingly.

The authors qualitatively do this for Test A in Table 2, but they don't explicitly state the importance of evaluating all possible combinations of predictor variables. They also don't provide any quantitative weighting measure of all possible predictor combinations. For example, they state for Test A that RHi is the most important predictor for the presence of cirrus (line 484-485). However, an argument could be made that Naer500 is just as important considering the approximately similar successful prediction rates of RHi and Naer500 as single variable predictors for all cirrus.

The methodology also does not account for the fact that RHi is intimately related to w and T. Thus, RHi inherently contains a signal of T and w within its calculation, and the findings actually state something along the lines of "water vapor and T is a better predictor than T", which is to be expected. It would benefit greatly to compare parameters which are completely independent of each other (e.g., aerosols vs RHi+w+T or water vapor vs T).

Additional comments:

Line 16: "with stronger AIE from larger aerosols than smaller ones" this doesn't make sense, please rephrase

Line 25: "Cirrus clouds are one…surface of 20%"

Line 28-30: Source?

Line 35-37: You're not using radiative effect correctly. Should change to flux.

Line 42-44: It seems findings suggest that either pore condensation freezing or depositional nucleation is occurring at very high rates in the upper troposphere given the strong relationships to Naer500 and cirrus properties. It would be helpful to circle back to this occasionally in the paper. I found myself wondering "why are we looking for ice nucleation signatures at temperatures where ice will instantaneously freeze?" while reading this paper. It seems this would be the reason why.

Line 44: The use of aerosol indirect effect is misleading in this paper. The term refers to how aerosols alter cloud properties by directly impacting cloud particle characteristics, which in turn feedback towards impacting radiative fluxes, precipitation initiation, etc. I'd recommend rewriting the paper and interchanging AIE with aerosol-cloud interactions or something comparable.

Line 51: $0 < T < 38C$.

Line 70: "allow for a"

Line 115: Perhaps reference the maps in the supplementary material only showing flight paths at T<-40C here.

Line 156: "were taken from the"

Line 165: "diameter" not "dynamic"

Line 187-188: Rewrite this so the brown and francis 1995 study is referring only OAP measurements.

Line 192: Rewrite this so it states what is being shown in the supplementary material not in the following sentence.

Line 201: Here and Line 280, you say the "delta-delta" method remove the "temperature effect". What does this mean? Shouldn't this be rephrased to say something along the lines of "detrending the data in relation to variable X"?

Line 205-206: Rephrase this sentence.

Section 2.2 seems out of place, and was confusing without giving a clear understanding of what machine learning will be used for. The objectives should be stated more clearly, and how the ML method specifically addresses them.

Line 234-237: This was confusing, until thinking about it you meant that quiescent conditions correspond to convective and non-quiescent correspond to in situ, yes? Either way, specifically state and argue for why your conditions are comparable to those from Kramer.

Line 251-253: 1) I think you mean Figure 2.

Line 229 & Line 260: These two range of +/- 30 seconds and 10 km, what is the rationale for selecting these lengths?

Line 281-283: How?

Line 288-290: How do we know it's sustained? I'd change "indicating" to "suggests"

Figure 6 caption: please specify that rows correspond to temperature, dRHi and w

Figure 6: Make sure x-axes ranges are consistent with left most columns and the right most column

Line 343-345: You can test for this right? Compare it with Naer500.

Figure 7 caption: think you labelled the red and green columns incorrectly

Line 380-381: Careful, you're not getting information of cloud formation here. Should rephrase to "occurrence"

Line 453-455: It's difficult to visualize this. Perhaps provide some quantitative measure to confirm this?

Line 495-496: I'm unsure what this means. If this is indeed what you meant, source?

Line 506-507: Did you flip "necessary" and "sufficient"?

---

## Referee Comment (RC2)

General Comments:

This is a very extensive observational study of cirrus clouds having near-global coverage, based on aircraft flights funded through NSF (7 campaigns) and NASA (5 campaigns). In particular, the smallest ice particles of the ice particle size distribution (PSD) are sampled, down to 1 μm, thus providing more useful information regarding the ice formation pathways (i.e., heterogeneous vs. homogeneous ice nucleation; henceforth het and hom). These ice PSD measurements are unique in that they have complementary measurements of relative humidity (RHi), aerosol particle PSDs, and vertical velocities (w). These complementary measurements are related to the ice PSD properties of ice water content (IWC), mean maximum dimension (Di), and number concentration (Ni) using a delta-delta method that correlates their fluctuations with those of the complementary measurements. Lastly, machine learning techniques are applied to better understand how IWC is affected by the complementary measurements.

I share the concerns expressed by Reviewer 1 regarding the practice of using in-cloud aerosol measurements since Na(500) (i.e., aerosol concentration between 0.5 μm and 1.0 μm) may be mostly small ice crystals. Why should fluctuations in Na(500) be so strongly correlated with fluctuations in IWC? What plausible physical process would explain these strong correlations? Using aerosol measurements just below cloud base could remedy this concern. Alternatively, if it could be shown that $N_i$(1-3μm) (i.e., the ice crystal number concentration between 1 and 3 microns as measured by the FCDP) is orders of magnitude less than Na(500), then it might be argued that ice crystals are a minor component of Na(500).

Our recent research shows that IWC and Ni track each other very closely when hom occurs. If Na for D > 0.5 μm here is strongly affected by Ni (as suggested by Reviewer 1), then these strong correlations may be partly due to fluctuations in hom (associated with higher IWC) correlated with fluctuations in Ni (associated with hom). Alternatively, one could argue that ISSRs (ice supersaturated regions) are common with higher INP (ice nucleating particle) concentrations impacting the RHi within these ISSRs to various degrees, resulting in correlations between IWC and Na(500) fluctuations. Such physical interpretations of these results are needed, even if they are only working hypotheses.

Specific Comments:

1. Lines 25 – 26: I don't see the justification for this statement (cirrus coverage of 20% to 40%). Sassen et al. (2009) estimates that global coverage for cirrus is 17%, while in Mace and Wrenn (2013), I could not find any mention of coverage.

2. Lines 48 – 50: Please support this statement with a reference. Cziczo et al. 2013 seems appropriate.

3. Line 107: Jensen et al. 2017b is cited for POSIDON but POSIDON is never mentioned in that article, which is concerned only with ATTREX. A good POSIDON reference is Schoeberl et al. (2019, JGR).

4. Lines 129 – 131: These PSD properties are defined in the abstract but not the text. Is that consistent with ACP policy? Also, Di is defined as number-weighted mean diameter, but diameter applies only to spheres. Please provide a more accurate definition, such as mean maximum dimension.

5. Lines 171 – 182: Regarding the NSF data, the Fast-2DC has a physical measurement range from 62.5 um to 1600 um, with 25 um bin widths (as stated here). Throwing out the first 3 bins would then limit the sampling range to 137.5 um to 1600 um. The measurement range of the CDP is from 1 to 50 um. This leaves a 87.5 um gap between 50 and 137.5 um. How is this gap addressed?

6. Line 247 – 250: At the bottom of Sect. 5.1.1 in Kramer et al. (2020) is the statement: "Because of the dangerous nature of measurements under such conditions, the frequency of convective – and also orographic wave cirrus – is underrepresented in the entire in situ climatology." Could this be an issue in this dataset as well? Moreover, in Sect. 4.2.2 of this article, we find "the higher (Ni) values at warmer temperatures in the Krämer et al. (2009) data set (Fig. S6, Supplement) were caused by flights where lee wave cirrus behind the Norwegian mountains were probed". To summarize, higher Ni values are associated with orographic gravity wave (OGW) cirrus clouds, and OGW cirrus are characterized by higher updrafts, more conducive to hom. In the satellite remote sensing studies of Gryspeerdt et al. (2018) and Mitchell et al. (2018), there are large regions of elevated Ni over and downwind of mountain barriers. Figure 1 of this paper does not show much sampling of such regions. Is it fair to say that OGW cirrus may have been under-sampled in these datasets leading to an underestimation of hom in the midlatitudes and polar regions? If so, please indicate this in the article.

7. Lines 272-275: Perhaps it is worth mentioning that the NSF data exhibits median Ni values near log(1.5), or 32 L-1, which is similar to median Ni in Kramer et al. (2020).

8. Lines 436 – 441: After studying Table 3, the "take-home message" for me is the following: At scales of 50-s or greater, dT + dRHi appears to be the most influential IWC predictor for quiescent cirrus. The 250-s scale appears to be the best IWC

predictor for non-quiescent cirrus (regarding dT + dRHi), with dlogNa(500) also having an impact in addition to dT + dRHi.  Should something along these lines be stated here?

9.  Lines 444 – 448:  For quiescent cirrus, it is true that IWC peaks ~ 110% RHi, but for non-quiescent cirrus, IWC peaks for RHi > 150%.  Please consider mentioning this important finding.

10. Lines 455 – 459:  In Fig. 10, what is responsible for the differences between panels a-b-c in the 1st row and panels d-e-f in the 2nd row?

11. Lines 513 -514: For a broader perspective, please mention the cirrus climatology of Kramer et al. (2020) and the number of field campaigns employed and their latitudes, surface types (land vs. ocean), and other relevant factors.

12. Lines 514 – 516:  As mentioned, the properties of orographic gravity wave (OGW) cirrus may differ considerably from other cirrus clouds and be widespread in coverage as noted in Joos et al. (2008, JGR), Barahona et al. (2017, Nature), Kramer et al. (2020, ACP), Mitchell et al. (2018, ACP), Gryspeerdt et al. (2018, ACP), Lyu et al. (2023, JGR) and other studies.  In Kramer et al. (2020), it was stated that OGW cirrus clouds were under-sampled due to dangerous flight conditions resulting from the higher updrafts.  The same is likely true of this study.  Please mention here this need to sample OGW cirrus clouds.

Technical Comments:

1.  Lines 159 – 161: The FCAS, being an aerosol probe, must have a measurement range of 70 - 1000 nm, not microns.

2.  Line 433:  Possible typo: 50 km => 250 km?